# In-situ preservation of nitrogen-bearing organics in Noachian Martian carbonates

Mizuho Koike [1✉], Ryoichi Nakada [2], Iori Kajitani [1,3], Tomohiro Usui [1,4], Yusuke Tamenori [5], Haruna Sugahara [1] & Atsuko Kobayashi[4,6]

Understanding the origin of organic material on Mars is a major issue in modern planetary science. Recent robotic exploration of Martian sedimentary rocks and laboratory analyses of Martian meteorites have both reported plausible indigenous organic components. However, little is known about their origin, evolution, and preservation. Here we report that 4-billion-year-old (Ga) carbonates in Martian meteorite, Allan Hills 84001, preserve indigenous nitrogen(N)-bearing organics by developing a new technique for high-spatial resolution in situ N-chemical speciation. The organic materials were synthesized locally and/or delivered meteoritically on Mars during Noachian age. The carbonates, alteration minerals from the Martian near-surface aqueous fluid, trapped and kept the organic materials intact over long geological times. This presence of N-bearing compounds requires abiotic or possibly biotic N-fixation and ammonia storage, suggesting that early Mars had a less oxidizing environment than today.

[1] Department of Solar System Sciences, Institute of Space and Astronautical Science, Japan Aerospace Exploration Agency, 3-1-1 Yoshinodai, Chuo-ku, Sagamihara, Kanagawa 252-5210, Japan. [2] Kochi Institute for Core Sample Research, Japan Agency for Marine-Earth Science and Technology (JAMSTEC), 200 Monobe, Nankoku, Kochi 783-8502, Japan. [3] Department of Earth and Planetary Science, The University of Tokyo, 7-3-1 Hongo, Bunkyo-ku, Tokyo 113-0033, Japan. [4] Earth-Life Science Institute, Tokyo Institute of Technology, 2-12-1 Ookayama, Meguro, Tokyo 152-8550, Japan. [5] Spectroscopy and Imaging Division, Japan Synchrotron Radiation Research Institute, 1-1-1 Koto, Sayo-cho, Sayo-gun, Hyogo 679-5198, Japan. [6] Division of Geological and Planetary Sciences, California Institute of Technology, Pasadena, CA 91125, USA. ✉email: koike@planeta.sci.isas.jaxa.jp

Questions concerning life on Mars have driven intensive studies of the Red Planet for decades, including focused investigation of possible organic molecules in recent Mars exploration. NASA's Mars Science Laboratory, Curiosity, reported various organic materials including sulfur and/or chlorine-bearing hydrocarbons from ~3.5 Ga mudstones in Gale crater[1–5]. Cl-bearing methane was also found from an earlier exploration by the Viking landers, which was considered to be of Martian origin[6]. These previous investigations suggested the existence of organic matter in the near-surface system on Mars. However, little is known about the origin, distribution, preservation, and evolution of such organics, as well as their possible relationship with Martian biological activity.

Along with robotic exploration, complementary knowledge has been obtained from detailed geochemical investigations of Martian meteorites[7]. Allan Hills (ALH) 84001, a unique early/middle Noachian igneous rock[8–10], contains fine-grained assemblages of secondary carbonate minerals[11–20] (Fig. 1; hereafter referred to as ALH carbonates). Previous studies reported the presence of organic carbon components in ALH carbonates, which are either

Martian or terrestrial contaminants[21–27]. Because ALH carbonates are considered to have precipitated via a low-temperature aqueous alteration at 4.04–3.90 Ga at Martian near-surface[18,19], their organic records, if any, should reflect the geochemical conditions at Noachian Mars. In situ study of their chemical speciation, including N, H, O, and S, will help further understanding.

Nitrogen (N) is an essential element for all life on Earth, as it is necessary for protein, DNA, RNA, and other vital materials. Nitrogen is also a useful geochemical tracer to reveal the co-evolution of the planetary atmospheres, hydrospheres, lithospheres, and biospheres. Previous studies of Martian N-chemical and isotopic signatures were done by bulk-rock destructive analyses[23,28–30]. However, in situ analysis of nitrogen chemical speciation has not been achieved due to technical difficulties. In this study we demonstrated for the first time the presence of trapped N-bearing organic compounds from micrometer-scale in situ analysis of N K-edge micro X-ray absorption near-edge structure (μ-XANES) on the 4 Ga ALH carbonates. Several carbonate grains were peeled off from a rock fragment of ALH 84001, 248 (allocated from NASA JSC in Texas, US), using silver double-sided sticky tape (Fig. 1b, d, e), which allowed us to investigate the interiors of the individual carbonate grains. Similarly, a silicate grain in the same rock fragment was collected to serve as a background control (Fig. 1c, f). Nitrogen μ-XANES measurements were conducted along with various N-bearing reference compounds at the SPring-8 synchrotron facility at Hyogo, Japan. The XANES spectra indicate the presence of indigenous N-bearing organics, which may have been trapped in ALH carbonates on Noachian Mars and preserved since then.

## Results and discussion

**Detection of nitrogen-bearing organics in ALH carbonates.** Nitrogen XANES spectra of ALH carbonates present two prominent absorption peaks at 398.9 and 399.9 eV with a broader absorption peak around 408 eV (Fig. 2). Their detailed analytical conditions are described in the "Method" section and Supplementary Table 1. There are additional smaller peak(s) between 400.7 and 402 eV. Such spectral shapes do not match the XANES spectra of molecular nitrogen ($N_2$), sodium nitrate ($NaNO_3$), or ammonium chloride ($NH_4Cl$)[31,32], suggesting that contributions from the inorganic N-bearing species are insignificant. On the other hand, the first two peaks at 398.9 and 399.9 eV are consistent with the absorptions of organic imino and nitrile groups, respectively (Supplementary Table 2). Pyridinic N-heterocyclic groups also have a similar energy range[31–33]. The third peak(s) between 400.7 and 402 eV may correspond to pyrrolic N-heterocyclic, amide, and/or amino groups[31,33]. The all XANES spectra are provided in the Supplementary Notes and Supplementary Figs. 1–3. Their original data are available in Supplementary Table 3. It is uncertain whether all of these organic groups are intrinsic to ALH carbonates or if some of them are due to X-ray beam damage during the XANES measurements. Considering that the imino and/or nitrile signatures were also observed in our amino acid references, some of the imino and/or nitrile features may come from the X-ray-induced decomposition of the intrinsic compounds (Supplementary Fig. 4). Consequently, our XANES spectra indicate that ALH carbonates contain a variety of the N-bearing organic components, whereas contributions from the inorganic N (e.g., $N_2$, nitrate or ammonium salt) are negligible. Plausible organic groups are the imino, nitrile, N-heterocyclic, amide, and/or amino groups.

**Examination of the possible Martian records.** When analyzing extraterrestrial organic materials, terrestrial contamination is

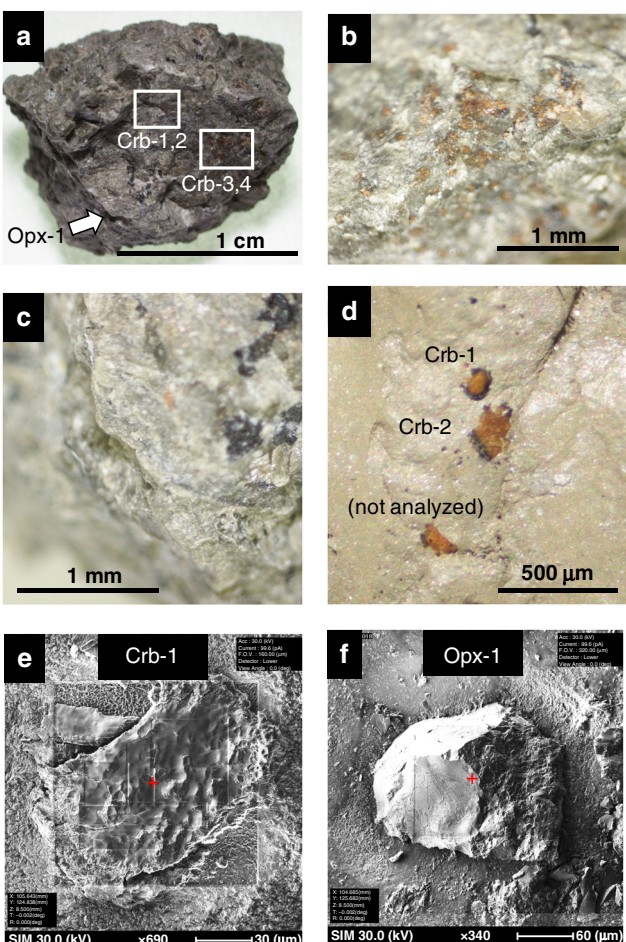

**Fig. 1 Optical and secondary electron images of ALH carbonates. a** Rock fragment of ALH 84001, 248. The whole size is ~1.5 cm. White squares (Crb-1,2 and Crb-3,4) indicate the locations of orange-colored carbonate patches used for our XANES measurements. A silicate grain was collected from the visibly carbonate-free area, shown by a white arrow in this image (Opx-1). Enlarged images of the carbonate patches (**b**) and the carbonate-free silicates (**c**). **d** The carbonate grains on a silver double-sided sticky tape. Secondary electron images of the ALH carbonate (**e**) and silicate (**f**) grains. These images were taken after the SEM-FIB processing to remove surface contaminants (see "Methods").

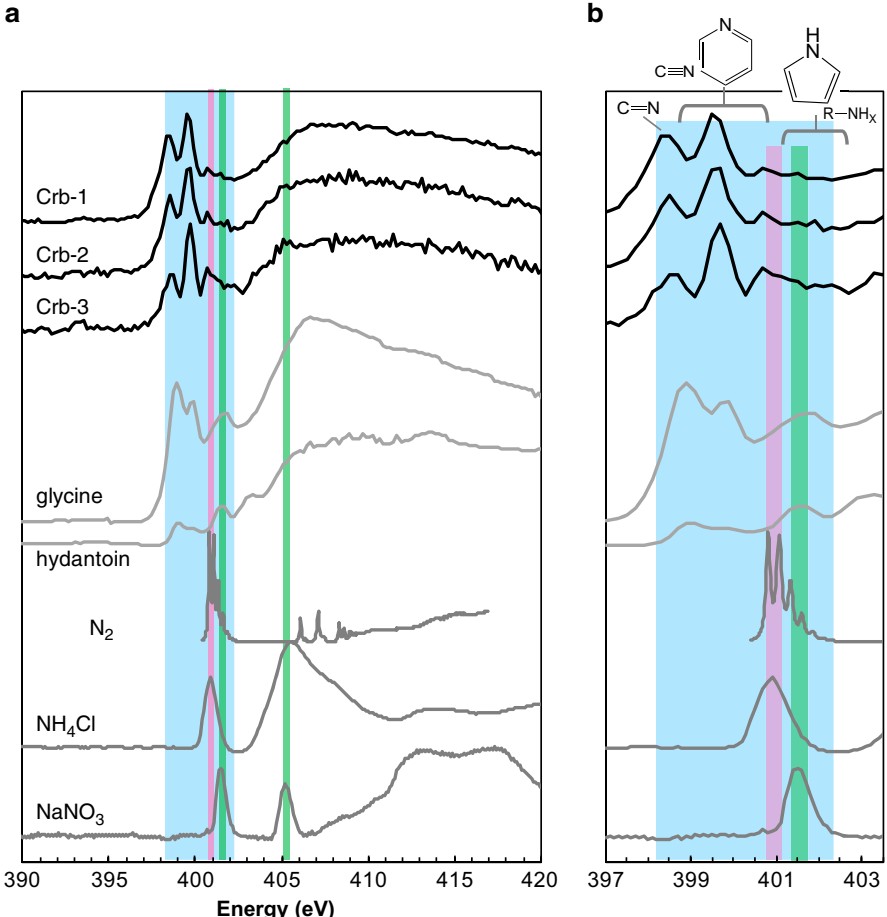

**Fig. 2 Nitrogen K-edge XANES spectra. a** A whole spectral image. The upper three (Crb-1 to Crb-3) are ALH carbonates, the others are the selected reference compounds. **b** An enlarged area of the energy between 397 and 403.5 eV. Significant absorption peaks for $N_2$ (400.8 eV; magenta), $NaNO_3$ (401.4 and 405.2 eV; green), and the organic compounds (398–402.5 eV; light blue) are highlighted in both images. Major N-bearing groups of imino, nitrile, N-heterocyclic, and amino groups in this energy range are plotted. The other XANES spectra obtained in this study are shown in Supplementary Figs 1–3.

always a serious concern. In this study, the possibility of laboratory contamination is minimized by conducting experiments carefully in a class 100 clean lab (for detail, see "Method" and Supplementary Discussion). On the other hand, contamination may have occurred from Antarctic ice before the collection of this meteorite. According to the stepwise heating analyses and the laser desorption mass spectrometry analyses for carbon isotopic ratios of ALH 84001[24,25], some portions of its non-carbonates C might be of Martian origin, whereas the other parts could be terrestrial contaminants. The presence of amino acids in ALH carbonates was reported through the bulk destructive analysis of high-performance liquid chromatography[23], most of which were regarded as Antarctic contaminants. At this moment, it is difficult to extract the Martian indigenous components from the mixtures with terrestrial contaminants. However, our in situ μ-XANES analyses reduce the risk and degree of contamination by focusing on the apparently fresh interior of the carbonate grains.

Detection limits of XANES measurements are typically in the order of 10 s of ppm to a few ppm[34]. Although this study does not focus on quantitative analyses, ALH carbonates should contain N-bearing organics above such a level. In contrast, concentrations of the organic materials in Antarctic surface ice are much lower[23], with the possible exception of locally enhanced areas in water gaps between ice crystals. Assuming sub-ppb-levels for the organic concentration in Antarctic ice meltwater co-existing with

ALH 84001, intense incorporation of the organic matter into the interior of ALH carbonates as high as >1000 times would be required for the XANES detection (e.g., ppm-level). Such a concentration is difficult to achieve via simple adsorption because partition coefficients between minerals and organics in the co-existing liquids are far below 1000[35,36]. Moreover, the apparent absence of nitrate in our ALH carbonates spectra (Fig. 2) seems to be incompatible with the contributions of Antarctic ice water or Martian oxidized nitrogen (discussed later). Based on the discussion above, the detected N-bearing organics in ALH carbonates are most likely of Martian origin.

**Conditions for long-term preservation of Martian organics.** If the organics in ALH carbonates are of Martian origin, they should have survived in the Martian near-surface system from Noachian period. The 4 Ga aqueous fluids, from which ALH carbonates precipitated, must have provided non-destructive environments for organic matter (i.e., moderate pH, Eh, UV, and cosmic-ray irradiation). Strong oxidants, such as chlorine oxides ($ClO_x^-$) and nitrogen oxides ($NO_x^-$), which are known to degrade organics and oxidize the Martian surface[37], have been reported in both current Martian regolith and in Amazonian-aged young Martian meteorites[30,38–40]. In contrast, our ALH carbonates do not show the XANES features of nitrate. Previous studies of ALH carbonates[41,42] propose less oxidizing conditions

for the 4 Ga Martian near-surface fluid (pH = 6–9 and Eh = −0.25–0 V). Under such an Eh-pH range, the co-existing nitrogen could be in the form of $N_2(aq)$ and/or $NH_4^+$ (Supplementary Fig. 5), consistent with the lack of the nitrate signatures. It is inferred that past Mars had the less oxidizing surface environments compared with today, at least locally and temporally. The organic matter was able to survive in the 4 Ga fluids, which were trapped and preserved in the hydrous alteration minerals (i.e., ALH carbonates) over long geological times. An experimental study demonstrates that amino acids can survive for tens of million years under the current Martian UV level, if they are embedded in the shallow regolith[43]. Energetic particles from galactic cosmic rays can penetrate down to 1–2 m below on the present Martian surface and would degrade organic molecules. However, theoretical study demonstrated that simple organic molecules (~100 amu) can survive at ~10 cm depth and complex organic molecules also have possibility to survive in much deeper subsurface for long period[44]. The host rock of ALH 84001 is likely to have resided in the subterranean system, indicating that its organic compounds have been protected from the severe UV and cosmic-ray irradiation for billions of years.

ALH 84001 has a complicated history of formation and metamorphism[8,9,11–20]. Its igneous crystallization age was dated at 4.09 ± 0.03 Ga using Lu–Hf chronology[9], whereas its Sm–Nd and Rb–Sr ages indicate 4.5 Ga[8], possibly due to counterclockwise rotation of the radiometric systems during later metamorphism. The K–Ar (Ar–Ar) system recorded a slightly younger age at ~3.9–4.1 Ga, associated with severe impact reheating[13]. ALH carbonates formed at the coincident timing. Their U–Pb and Rb–Sr chronologies revealed the crystallization age at 4.04–3.90 Ga[18].

After the formation of ALH carbonates, some of them suffered additional impact(s), being fractured and possibly reheated[20]. Magnesite-rich rims, typical characters for ALH carbonates, have been attributed either to the result of heat decomposition from earlier Fe-rich carbonates[13,16] or to the result of low-temperature fluid variation during precipitation[14,17]. In the former case, the reheating temperature would have had to be uniformly around 500 °C, which would have degraded considerable part of organic compounds[45,46]. However, clumped O and C isotopes in the carbonate yield the crystallization temperature of 18 ± 4 °C[19], which would have been reset to higher values even by brief heating to temperatures above 450 °C[47]. Such heating would also have erased the fine-scale geochemical zoning and would have remagnetized the rock. According to high-resolution magnetic studies[12,15], ALH 84001 passes a paleomagnetic conglomerate test, demonstrating that the interior was never above 40 °C from the time of ALH carbonates formation on Mars to the meteorite's arrival to Earth. Hence, our identification of the intact-N-bearing organic compounds trapped within ALH carbonates supports this latter interpretation.

**Origin of the nitrogen-bearing organics on Noachian Mars.** Possible origins for the Martian organics are in situ syntheses and/or meteoritical supplies (Fig. 3). As atmospheric dinitrogen ($N_2$) is chemically inert due to the high activation energy of its N≡N bond[48], fixation of N into accessible forms is required to produce the N-bearing compounds. Oxidized nitrogen (e.g., nitrate and nitrite) may be produced on Mars through thermal shocks by previous volcanic lightning, meteoritical impacts, cosmic-ray and solar X-ray irradiations[39,49,50]. Meanwhile, reduced nitrogen (e.g., ammonia and hydrogen cyanide) has not been identified on Mars, mainly because of their instability on the present Martian surface environments[39]. However, several abiotic paths for the reduced N are proposed for the Hadean Earth[51], such as the reduction of $NO_x$ by metallic iron and/or aqueous

solutions, as well as the photochemical reduction of atmospheric $N_2$. The similar processes could have occurred on early Mars. The less oxidizing conditions of the Noachian period, as discussed before, seem to be suitable for $NH_4^+$ (Supplementary Fig. 5). Moreover, the redox state of the Martian mantle is highly reducing with the $fO_2$ level at the iron-wustite buffer or slightly higher[52], where nitrogen should be present as ammonia[53]. The reduced N may have been partly supplied from the Martian interior through early volcanic events. Ammonia is a highly reactive and important starting chemical for producing more complex N-bearing molecules. ALH carbonates may have recorded a part of the N-bearing organic/inorganic chemical evolutions process of Noachian Mars.

Meteoritic supply of organic compounds is also a plausible interpretation. Carbonaceous chondrites are known to contain a variety of insoluble and soluble N-bearing organic groups including amino acids, amines, amides, N-heterocyclic compounds, and macromolecular organic matter (e.g., polycyclic aromatic hydrocarbons) with sub-ppm to 100 s ppm levels[54]. Interplanetary dust particles and comets contain various components as well. Previous detection of the organic matter in a much younger Martian meteorite (i.e., Tissint; with a crystalization age of ~570 Myr) may reflect such an extra-Martian origin[40]. At 4 Ga, however, the higher impact flux of meteoritical materials should have been supplied larger quantities of organic material onto Earth and possibly onto Mars[55,56], although their influx rates are not determined quantitatively.

Whatever the origin, the presence of the organic and reduced nitrogen on early/middle Noachian Mars indicates the importance of Martian nitrogen cycle. If considerable amounts and variations of organic matter were produced and/or delivered and preserved at the Martian near-surface system over geological time scales, these compounds have a chance to evolve into more complicated forms. It is expected that additional hidden records of the Martian nitrogen cycle will be acquired by future investigations, including a sample return mission from the Martian Moons[57] (Martian Moons eXploration), Mars Sample Return missions, and exploration of the Martian subsurface, as well as further advanced studies of Martian meteorites.

## Methods

**Sample preparations.** A rock fragment from the interior of Allan Hills 84001, sub-sample 248, was donated by the Meteoritical Working Group, NASA[58]. The fragment was carefully observed under an optical microscope in a class 100 clean room at the Earth-Life Science Institute, Tokyo Institute of Technology, Japan. Numerous patches of characteristic orange-colored carbonates were observed on the apparently fresh surface of this fragment (Fig. 1). The carbonates were fragile and were able to be peeled off using a silver double-sticky tape (Nisshin EM Co., Ltd). To avoid laboratory contamination, none of the organic materials normally utilized were used for the preparation of a thin rock section (e.g., epoxy resin, glue, and polishing paste), except for the silver tape. In addition to the carbonates, small grains of ALH silicate (orthopyroxene grains) were collected from the area without any visible carbonate blebs in the same rock fragment. Because the igneous silicate minerals in ALH 84001 are not expected to contain indigenous organic matter, the results of ALH silicate can be used to estimate the background level of our N μ-XANES measurement.

The plucked carbonates and silicate grains were observed using a Focused Ion and Electron Beam system (FIB-SEM; NX 2000) at the Extraterrestrial Sample Curation Center, Japan Aerospace Exploration Agency, Japan. Four grains of the ALH carbonates (Crb-1 to Crb-4) and a single grain of the ALH silicate (Opx-1), with sizes of ~50–200 μm in diameter, were selected for the following XANES analyses. In order to reduce the possibility of contamination, the surfaces of the ALH carbonates and silicates were etched to ~1 μm depth using a 12 nA–100 pA $Ga^+$ ion beam with an acceleration voltage at 30 kV on the FIB-SEM.

**Nitrogen XANES measurements.** Nitrogen K-edge μ-XANES spectra were measured at BL27SU of SPring-8 synchrotron facility in Hyogo, Japan in December 2018 and April 2019. Radiation from an undulator was dispersed by a soft X-ray monochromator with varied-line-spacing plane grating. The photon energy resolution during the measurements was set at 40 meV. The X-ray energy of the two periods was identical as checked by a boron nitride powder, which were calibrated

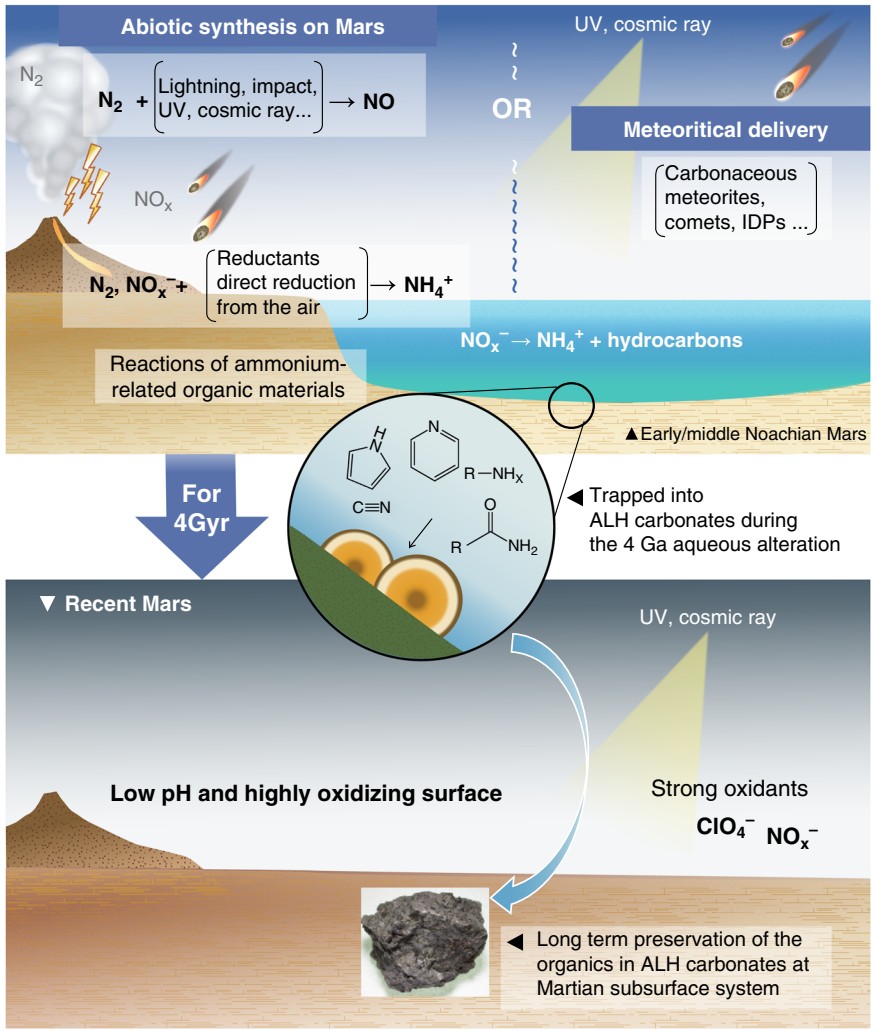

**Fig. 3 Possible history for the Martian N-bearing organics.** Nitrogen-bearing organic matter was either synthesized locally or delivered meteoritically on the early Mars. The former case requires an abiotic reduction of N (e.g., $N_2$, $NO_x \rightarrow NH_3$) to start the ammonia-related chemical reactions. The latter case is also possible if adequate amounts of N-bearing components were supplied. This organic matter survived in the 4 Ga Martian (near)surface fluids and was trapped into ALH carbonates during precipitation. The carbonates resided in the subterranean system, preserving the organic components over long geological times.

with the peak maximum of $N_2$ gas at 400.8 eV. The XANES spectra obtained during December 2018 were of better quality than those of April 2019 because the intensity of the incident X-ray beam measured at the former period was higher by ~10 times than the latter. To obtain the reference XANES spectra, two species of N-bearing inorganic reagents; i.e., sodium nitrate ($NaNO_3$) and ammonium chloride ($NH_4Cl$) (both from Fujifilm Wako Pure Chemical Corp.), as well as dinitrogen ($N_2$) gas in air were measured in the total electron yield (TEY) mode. In addition, the reference XANES spectra of several N-bearing organic compounds were obtained; i.e., amino acids (glycine: $C_2H_4NO_2$, DL-alanine: $C_3H_7NO_2$, 2-aminoisobutyric acid: $C_4H_9NO_2$, and DL-aspartic acid: $C_4H_7NO_2$), imino acid (sarcosine: $C_3H_7NO_2$), N-heterocycles (hydantoin: $C_3H_4N_2O_2$, DL-pyroglutamic acid: $C_5H_7NO_2$), and amine (methylamine hydrochloride: $CH_5N \cdot HCl$). Hydantoin, DL-aspartic acid, sarcosine, and methylamine hydrochloride was measured in fluorescent yield (FY) mode using a single element silicon drift detector (Amptek), whereas all the others were measured in TEY mode. We also measured three kinds of scandium-bearing chemical reagents; i.e., scandium oxide ($Sc_2O_3$; from Fujifilm Wako) and scandium carbonate hydrate ($Sc_2(CO_3)_3 \cdot H_2O$, from Alfa Aesar, Fisher Scientific) in TEY mode, and scandium chloride hexahydrate ($ScCl \cdot 6H_2O$; from Fujifilm Wako) in FY mode, because $L_{II}$-edge and $L_{III}$-edge absorption energies of Sc locates in the same energy range of the N K-edge in XANES. The XANES spectra of the carbonates and silicate grains from ALH 84001 (Crb-1 to 4 and Opx-1), were measured in FY mode and compared with the reference N and Sc XANES spectra. We also measured a ~1 × 1 mm piece of the unused silver tape in FY mode in order to check for the possibility of experimental contamination. Two grains of the ALH carbonates (Crb-1 and 2) were measured twice, both on December 2018 and April 2019. Supplementary Table 1 summarizes the experimental conditions of all the samples and reference materials analyzed in this study.

An incident X-ray beam was focused on the sample surface with a spot size of 10 μm (V) × 200 μm (H). All the reference materials, which were in powdered condition and a piece of the silver tape were measured using this wide beam. We further used a focused narrow beam for the measurements of the small grains of ALH carbonates and silicate (Fig. 1). This narrow beam was achieved with a poly-capillary lens to focus on the final spot size of ca. 25 μm (diameter) in December 2018, and by closing a slit to achieve the spot size of 10 μm (V) × 30 μm (H) in April 2019. The incident beam intensity of this cutoff beam was reduced to ~1/10 of the original beam. One grain of the ALH carbonates (Crb-1) was measured with both the narrow beam (December 2018 and April 2019) and the wide beam (April 2019), to compare the effects of the different beam intensities. Because the radiation of a strong X-ray may cause severe damage over the whole carbonate grain, we measured the wide-beam mode after all other measurements were completed.

All measurements were performed under ambient temperature and high vacuum (~5 × 10$^{-4}$ Pa for December 2018 and ~1 × 10$^{-5}$ Pa for April 2019). The incident X-ray energy range was 385–425 eV with energy steps mainly of 0.2 eV. All the samples (i.e., ALH carbonates and silicate grains) were measured with the integration of 10–15 s per energy step with repeated scans of 2–10 times, which means the total integration time was 30–150 s/step (see Supplementary Table 1). On the other hand, all the reference materials were measured in 1–15 s/step without repeated scans. The silver tape was measured in 15 s/step. Detailed measurement conditions are summarized in Supplementary Table 1. Prior to these μ-XANES measurements, the X-ray fluorescence (XRF) mappings of Si, Al, Mg, Fe, O, and C of ALH carbonates were obtained by scanning at 40–200 μm per step at incidence photon energy of 1900 eV to locate the targets. For μ-XANES analysis, full XRF spectral data were obtained for each point of excitation energy and the element-selected fluorescence yield data were extracted from the XRF data set.

## Data availability

All spectral data analyzed during this study are provided as figures (Fig. 2 in the main text and Supplementary Figs. 1–3 in our Supplementary Information). Their original data are available in Supplementary Table 3.

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

## Acknowledgements

A rock fragment of ALH 84001, 248 was kindly donated by the Meteoritical Working Group, Johnson Space Center, NASA. The FIB-SEM-based sample preparation was performed with the kind approval of Astromaterials Science Research Group (ASRG), JAXA. The synchrotron radiation experiments were performed with the approval of the Japan Synchrotron Radiation Research Institute (JASRI) (Proposal No. 2018B1062 and 2019A1389). This study is partly supported by JSPS Grants-in-Aid for Scientific Research (KAKENHI) to M. K. (18J02005 and 19K14790), to R. N. (17H06458), to T. U. (19H01960, 17H06459, 15KK0153, and 16H04073), and to A. K. (26630062 and 16H04276). A. K. was also supported by USA-DARPA Grant D17AC00019. We thank Prof. J. L. Kirschvink of the Division of Geological and Planetary Sciences, California Institute of Technology, for providing informative comments on an earlier draft of the manuscript.

## Author contributions

M. K., A. K., T. U., and H. S. prepared the samples and the reference materials. M. K., R. N., I. K., T. U., and Y. T. conducted the N μ-XANES measurements. M. K. took the lead in writing the manuscript. All the authors contributed in the discussion of the results and the improvement of the manuscript.

## Competing interests

The authors declare no competing interests.
