## [Peer Review File · Nature Communications]

Reviewers' comments:

Reviewer #1 (Remarks to the Author):

This manuscript reports the detection of organic N species in Mars meteorite ALH84001 by in situ μ -XANES analysis. Such a detection would have great significance for understanding whether there was ever a nitrogen cycle on Mars and contribute new information to the study of the habitability of early Noachian Mars. The manuscript is clearly written and adequately addresses the question of terrestrial contamination as the source of these N bearing organics. The argument about the concentrations required for XANES vs. ppb organic abundances of organics in Antarctic surface ice in lines 104-113 is fairly convincing. While I generally accept the case for Martian origin of these compounds, I cannot speak to the identification of N bearing organics by μ -XANES. This technique is not my area of expertise. The spectral features appear to match up, but I don't know how something like the cation of the nitrate would affect the spectral signature of nitrate. As long as someone else has vetted these identifications, the manuscript should be published with minor revision.

I note that ionizing radiation/GCRs (e.g., Pavlov et al. 2012) should be included in the preservation discussion, particularly l. 134-137.

Pavlov, A. A., et al. "Degradation of the organic molecules in the shallow subsurface of Mars due to irradiation by cosmic rays." *Geophysical research letters* 39.13 (2012).

Reviewer #2 (Remarks to the Author):

The manuscript by Koike et al. is an interesting, topical and potentially significant study of N-bearing compounds in the unique martian meteorite ALH84001. The form of N-bearing compounds is an essential part of astrobiological studies of Mars.

A key challenge for this type of work is establishing the absence of contamination and establishing that the trace N species signatures are truly specific to the secondary minerals. Koike et al. have managed this in the Supplementary Materials section giving confidence about their core XANES data.

I think it is ultimately publishable but some edits are necessary. Specifically, The ALH84001 carbonates experienced a series of events which hasn't really been described in this manuscript. Notably, it is generally thought that following low temperature precipitation there was a significant shock event related to mineralogical zonation and precipitation of Fe oxides.. You can see an up to date summary of the models for ALH84001 in Bridges J.C., Hicks L.J, Treiman A. (2019) Carbonates on Mars. In 'Volatiles on Mars'. 1st edition. Elsevier. Editors Filiberto and Schwenzer. pp 426. What effect will shock have had on the N speciation described here?

Although an age of 4.1 Gyr is cited in the paper, the chronometer should be given and a slightly fuller description of the complex series of events recorded by the radiometric dating.

Abstract line 31. Do you mean ' carbonates and hydrous, unheated minerals?'

Reviewer #3 (Remarks to the Author):

Review NCOMMS-19-38604-T, In-situ preservation of nitrogen-bearing organics in early Noachian Martian carbonates by Koike et al.

Interesting study using state-of-the-art synchrotron N K-edge μ -XANES to determine the presence and potential origin of N-compounds in a Martian meteorite. Results are of high quality and I feel that the analyses/interpretations are sound/convincing. The MS is well written and the outcomes/conclusions are novel and of interest to the reader of NCOMMS. I have no major comments and therefore suggest accepting the MS upon minor revision.

Below are a few minor comments/suggestions:

Line 23. Delete 'ancient'. In addition, I feel that in most cases that 'ancient' is used in the MS this is actually obsolete and encourage the authors to check the MS carefully.

Line 40. Delete 'have'.

Line 41. Delete 'present'. This since we have no idea how old these systems really are.

Line 50. Delete 'Have'.

Line 54. Nitrogen is not only an essential element for terrestrial organisms but for all life on earth.

Lines 59-61. Replace 'In this study, we accomplished the micrometer-scale in situ analysis of N K-edge micro X-ray absorption near-edge structure (μ -XANES) on the 4 Ga ALH carbonates for the first time' with 'In this study, for the first time, micrometer-scale in situ analysis of N K-edge micro X-ray absorption near-edge structure (μ -XANES) on the 4 Ga ALH carbonates were accomplished.'

Lines 66-67. Replace 'We conducted their N μ -XANES measurements along with various N-bearing reference compounds at the SPring-8 synchrotron facility (Hyogo, Japan). Our XANES spectra...' with 'N μ -XANES measurements were conducted along with various N-bearing reference compounds at the SPring-8 synchrotron facility (Hyogo, Japan). The XANES spectra...'

Line 92. Replace 'In this study, we minimized the possibility of laboratory contamination by..' with 'In this study, the possibility of laboratory contamination is minimized by ...'

Line 131. Delete 'then'.

Line 172. Replace 'We expect that additional...' with 'It is expected that additional...'

Lines 272-73. Replace 'We obtained a rock fragment from the interior of Allan Hills 84001, sub-sample 248, donated to us by the' with 'A rock fragment from the interior of Allan Hills 84001, sub-sample 248, was donated by the....'

Lines 278-279. Replace '..., we did not use any organic materials normally utilized for the preparation of a...' with '..., ion, none of the organic materials normally utilized were used for the preparation of a ...'.

Line 280-281. Replace '..., we collected small grains of ALH silicate (orthopyroxene grains)...' with '..., small grains of ALH silicate (orthopyroxene grains) were collected...'

Line 285. Replace 'We then observed the plucked carbonates and silicate grains using...' with 'The plucked carbonates and silicate grains were observed using....'

Lines 305-306. Replace '....we obtained the reference XANES spectra of several N

Reviewer #1 (Remarks to the Author):

This manuscript reports the detection of organic N species in Mars meteorite ALH84001 by in situ μ -XANES analysis. Such a detection would have great significance for understanding whether there was ever a nitrogen cycle on Mars and contribute new information to the study of the habitability of early Noachian Mars. The manuscript is clearly written and adequately addresses the question of terrestrial contamination as the source of these N bearing organics. The argument about the concentrations required for XANES vs. ppb organic abundances of organics in Antarctic surface ice in lines 104-113 is fairly convincing. While I generally accept the case for Martian origin of these compounds, I cannot speak to the identification of N bearing organics by μ -XANES. This technique is not my area of expertise. The spectral features appear to match up, but I don't know how something like the cation of the nitrate would affect the spectral signature of nitrate. As long as someone else has vetted these identifications, the manuscript should be published with minor revision.

I note that ionizing radiation/GCRs (e.g., Pavlov et al. 2012) should be included in the preservation discussion, particularly l. 134-137.

*Pavlov, A. A., et al. "Degradation of the organic molecules in the shallow subsurface of Mars due to irradiation by cosmic rays." *Geophysical research letters* 39.13 (2012).*

Response:

We now refer to Pavlov et al. (2012) in our preservation discussion as suggested. (Lines 133-140 and Reference #42 in the revised MS).

Under present Martian condition (thin atmosphere and no global magnetic field) the effect of galactic and solar cosmic ray irradiation is not negligible. Energetic particles from GCRs can generally penetrate 1–2 m below the surface of regolith, where the organic molecules may be degraded. However, this is only the case for complex organic molecules which are at near the surface. The calculation by Pavlov et al. demonstrates that simple organic molecules (≤ 100 amu) can survive at ~ 10 cm depth, and it is even possible for complex molecules to survive at greater depths for long periods of time. It is most likely that ALH carbonates preserved the N-bearing organic molecules in the Martian subterranean for a long period, shielding them from the severe cosmic ray irradiation (and from strong oxidants and UV irradiation as well).

Reviewer #2 (Remarks to the Author):

The manuscript by Koike et al. is an interesting, topical and potentially significant study of N-bearing compounds in the unique martian meteorite ALH84001. The form of N-bearing compounds is an essential part of astrobiological studies of Mars. A key challenge for this type of work is establishing the absence of contamination and establishing that the trace N species signatures are truly specific to the secondary minerals. Koike et al. have managed this in the Supplementary Materials section giving confidence about their core XANES data.

I think it is ultimately publishable but some edits are necessary. Specifically, The ALH84001 carbonates experienced a series of events which hasn't really been described in this manuscript. Notably, it is generally thought that following low temperature precipitation there was a significant shock event related to mineralogical zonation and precipitation of Fe oxides. You can see an up to date summary of the models for ALH84001 in Bridges J.C., Hicks L.J, Treiman A. (2019) Carbonates on Mars. In 'Volatiles on Mars'. 1st edition. Elsevier. Editors Filiberto and Schwenger. pp 426. What effect will shock have had on the N speciation described here?

Response:

We now refer to the review by Bridges et al. (2019) as suggested, as well as several additional key studies in the revised MS (Reference # of 11–15, 19), and discussed the probability of the shock-induced reheating of the measured carbonates (Lines 148-161 in the revised MS).

Bridges et al. (2019) provide a broad review of the distribution of carbonates on the present Martian surface and in Martian meteorites (ALH 84001 and nakhlites). In their discussion, they attribute the striking zonation feature of carbonate blebs ALH84001 to the heat decomposition of Fe-bearing carbonates to magnetite during a shock heating event. However, such interpretation ignores several important and highly-cited papers that demonstrate the bulk of this meteorite, and the magnetite-rimmed carbonates, could not have been raised to high enough temperatures to cause this. These are:

- (1) Kirschvink, et al., 1997. Paleomagnetic evidence of a low-temperature origin of carbonate in the Martian meteorite ALH84001. *Science* 275, 1629-1633.

This work reports that the mm-sized orthopyroxene clasts of ALH 84001, separated along the grain boundaries, pass a paleomagnetic conglomerate test. Their results demonstrate that the host rock was not heated high enough to re-magnetize the grains after they were put in place. Bulk reheating of ALH 84001 high enough to form the magnetite would make all of the

magnetic directions to point in the same direction, which would cause the paleomagnetic conglomerate test to fail.

- (2) Weiss et al., 2000. A low temperature transfer of ALH84001 from Mars to Earth. *Science* **290**, 791-795.

This work with scanning (SQUID) magnetic microscopy puts a numerical value on the peak (re)heating temperature at < 40 °C for the entire time since ALH carbonates formed, ejected from Mars, landed in Antarctica and even until the meteorite was brought to the Johnson Spaceflight Center in Houston. These magnetic studies confirm that the subsequent processes of ALH 84001 (e.g., impact, ejection from Mars, and arrival to Earth) did not heat ALH carbonates above 40°C. Moreover, the following isotopic studies support also their low temperature formation and preservation processes:

- (3) Halevy et al., 2011. Carbonates in the Martian meteorite Allan Hills 84001 formed at 18 ± 4 °C in a near-surface aqueous environment. *Proc. Natl. Acad. Sci.* **108**, 16895-16899.

The study by Halevy et al. showed that ALH carbonates formed at the low temperature of 14–22 °C via aqueous alteration, by measuring the abundance of the multiple heavy isotopes of C and O (i.e., “clumped” isotope thermometry).

- (4) Stolper, D. A., and Eiler, J. M., 2015. The kinetics of solid-state isotope-exchange reactions for clumped isotopes: A study of inorganic calcites and apatites from natural and experimental samples. *American J. Sci.* **315**, 363-411.

This study demonstrates that the clumped isotopic signature in the carbonates would be easily reset if they were briefly (re)heated above 450 °C.

All of the data from ALH84001 confirm that ALH carbonates did not experience high temperature heating since their crystallizations, although the host rock may have been suffered impact deformation event(s). The characteristic zoning of the carbonates is most likely the primary feature, i.e., this zoning formed in the fluctuations of the co-existing fluidal chemistry. Consequently, the post-crystallization process is not enough to effect the nitrogen speciation in ALH carbonates.

Although an age of 4.1 Gyr is cited in the paper, the chronometer should be given and a slightly fuller description of the complex series of events recorded by the radiometric dating.

Response: A summary of the formation and metamorphic history of ALH 84001 has been described in our discussion (Lines 141-147 in the revised MS), in addition to the brief introduction of ALH carbonates (Lines 45-51 in the revised MS) as suggested. In addition to the radiometric dating studies by Lapen et al. (Ref #9) and Borg et al. (Ref #17), we have referred the previous review of Martian meteorites' chronologies by Nyquist et al. (Ref #8).

Abstract line 31. Do you mean 'carbonates and hydrous, unheated minerals?'

Response: The original sentence might be inaccurate and misleading. We have replaced the phrases with 'alteration minerals from the Martian near-surface aqueous fluid' (Line 31)

Reviewer #3 (Remarks to the Author):

Interesting study using state-of-the-art synchrotron N K-edge μ -XANES to determine the presence and potential origin of N-compounds in a Martian meteorite. Results are of high quality and I feel that the analyses/interpretations are sound/convincing. The MS is well written and the outcomes/conclusions are novel and of interest to the reader of NCOMMS. I have no major comments and therefore suggest accepting the MS upon minor revision.

Below are a few minor comments/suggestions:

Line 23. Delete 'ancient'. In addition, I feel that in most cases that 'ancient' is used in the MS this is actually obsolete and encourage the authors to check the MS carefully.

Response: We have deleted the word as suggested. Also, we have also checked the whole MS and deleted or replaced 'ancient' from 8 sentences in total (Lines 23, 30, 32, 38, 128, 192, 195, and 293 in the revised MS)

Line 40. Delete 'have'.

Response: We have deleted the word as suggested.

Line 41. Delete 'present'. This since we have no idea how old these systems really are.

Response: We have deleted the word as suggested.

Line 50. Delete 'Have'.

Response: We have deleted the word as suggested. (Line 47 in the revised MS)

Line 54. Nitrogen is not only an essential element for terrestrial organisms but for all life on earth.

Response: We have revised the words as suggested. (Line 53 in the revised MS)

Lines 59-61. Replace 'In this study, we accomplished the micrometer-scale in situ analysis of N K-edge micro X-ray absorption near-edge structure (μ -XANES) on the 4 Ga ALH carbonates for the first time' with 'In this study, for the first time, micrometer-scale in situ analysis of N K-edge micro X-ray absorption near-edge structure (μ -XANES) on the 4 Ga ALH carbonates were accomplished.'

Response: We have revised the sentence as suggested. (Lines 58-60 in the revised MS)

Lines 66-67. Replace 'We conducted their N μ -XANES measurements along with various N-bearing reference compounds at the SPring-8 synchrotron facility (Hyogo, Japan). Our XANES spectra...' with 'N μ -XANES measurements were conducted along with various N-bearing reference compounds at the SPring-8 synchrotron facility (Hyogo, Japan). The XANES spectra...'

Response: We agree. We have revised the sentence as suggested. (Lines 65-66 in the revised MS)

Line 92. Replace 'In this study, we minimized the possibility of laboratory contamination by...' with 'In this study, the possibility of laboratory contamination is minimized by ...'

Response: We have revised the sentence as suggested. (Line 91 in the revised MS)

Line 131. Delete 'then'.

Response: We have deleted the word as suggested. (Line 130 in the revised MS)

Line 172. Replace 'We expect that additional...' with 'It is expected that additional....'.

Response: We have revised the sentence as suggested. (Line 196 in the revised MS)

Lines 272-73. Replace 'We obtained a rock fragment from the interior of Allan Hills 84001, sub-sample 248, donated to us by the ...' with 'A rock fragment from the interior of Allan Hills 84001, sub-sample 248, was donated by the....'

Response: We have revised the sentence as suggested. (Lines 297-298 in the revised MS)

Lines 278-279. Replace '..., we did not use any organic materials normally utilized for the preparation of a...' with '..., ion, none of the organic materials normally utilized were used for the preparation of a ...'

Response: We have revised the sentence as suggested. (Lines 303-304 in the revised MS)

Line 280-281. Replace '...., we collected small grains of ALH silicate (orthopyroxene grains)...' with '..., small grains of ALH silicate (orthopyroxene grains) were collected....'

Response: We have revised the sentence as suggested. (Lines 305-306 in the revised MS)

Line 285. Replace 'We then observed the plucked carbonates and silicate grains using...' with 'The plucked carbonates and silicate grains were observed using...'

Response: We have revised the sentence as suggested. (Line 310 in the revised MS)

REVIEWERS' COMMENTS:

Reviewer #1 (Remarks to the Author):

No further comments, it appears that all reviewers' comments have been addressed. Publication with no further revision recommended.

Reviewer #2 (Remarks to the Author):

I think the authors have given a reasonable amount of information about ALH84001 carbonate formation. More could be given e.g. about the likely rapid and metastable nature of those assemblages during initial precipitation from water (e.g. Bridges et al. 2001). However, I leave that at the discretion of the authors.

At several points the authors describe the assemblage as early Noachian. But ~4 Gyr is not usually considered early Noachian e.g. see Irwin, Tanaka et al. JGR 2013

Reviewer #3 (Remarks to the Author):

NA

A response file to review comments

Reviewer #2 (Remarks to the Author):

I think the authors have given a reasonable amount of information about ALH84001 carbonate formation. More could be given e.g. about the likely rapid and metastable nature of those assemblages during initial precipitation from water (e.g. Bridges et al. 2001). However, I leave that at the discretion of the authors.

At several points the authors describe the assemblage as early Noachian. But ~4 Gyr is not usually considered early Noachian e.g. see Irwin, Tanaka et al. JGR 2013

Response:

Although there are a number of important and detailed petrographic/geochemical studies concerning the formation of ALH 84001 carbonates, we consider that we have referred the adequate information here, as the main focus of this study is Martian nitrogen speciation.

According to the cratering chronology study by Hartmann and Neukum (2001) Space Sci. Rev. 96, 165-194, the age of ~4 Gyr is defined as Early to Middle Noachian. Irwin et al. (2013) JGR also refer to this definition. In our revised manuscript, we refer to Hartman and Neukum (2001) and removed the words 'early Noachian' from the text and the title. Instead, we use 'early/middle Noachian' or simply 'Noachian'.